# Early access to physiotherapy for infants with cerebral palsy: A retrospective chart review

**Linnéa Hekne[1], Cecilia Montgomery[2], Kine Johansen[2] ***

**1** Pediatric Department, Västmanland Hospital Västerås, Västerås, Sweden, **2** Department of Women's and Children's Health, Uppsala University, Uppsala, Sweden

* kine.johansen@kbh.uu.se

**Data Availability Statement:** Data cannot be shared publicly because it includes potentially identifying and sensitive patient information, therefore access is limited to by request by the

## Abstract

### Aim

This study aimed to investigate whether children with cerebral palsy (CP) had equal access to timely physiotherapy. Additionally, to learn more about clinical characteristics of infants with CP, we explored differences in neonatal clinical history and CP profile between children referred by a neonatologist or enrolled in neonatal follow-up and those referred by other healthcare professionals as well as those referred before and after 5 months corrected age.

### Methods

We conducted a retrospective chart review study including children born in Uppsala County, Sweden, from 2010 to 2016, who had received a CP diagnosis by July 2019. Entries by doctors and physiotherapists working at Uppsala University Children's Hospital were reviewed.

### Results

Thirty-eight children were included (21 girls, 55.3%) in the study. Twenty-two (57.9%) were born at term. Twenty-five children (66%) had their first visit to a physiotherapist before 5 months corrected age, and this included all children (n = 22, 57.9%) referred by a neonatologist or enrolled in neonatal follow-up. The latter group had significantly earlier access to physiotherapy compared to children referred by other healthcare professionals, with a median of 1.9 (min-max: -1-4) and 7.6 (min-max: 1–24) months, respectively (p < 0.0001). Referral source explained unique variance in predicting time of referral to physiotherapist ($R^2$ 0.550, B 4.213, p < 0.0001) when controlling for both number of risk factors and severity of motor impairment. However, number of risk factor was vital for early access to physiotherapy for children referred by other health care professionals.

Children referred by a neonatologist or enrolled in neonatal follow-up or referred before 5 months corrected age differed on all measured variables concerning neonatal clinical history and CP profile, compared to children referred by other healthcare professionals or after 5 months corrected age. The latter groups had milder forms of CP. In total, twenty-eight children (73.7%) were ambulatory at 2 years of age. Bilateral spastic CP was most common among those referred by a neonatologist or enrolled in neonatal follow-up or referred before

Regional Ethical Review Board in Uppsala, Sweden (reg. no. 2018/173). Data are available from Uppsala University, Legal Affairs Division (contact via e-mail: registrator@uu.se) for researchers who meet the criteria for access to confidential data.

**Funding:** The authors received no specific funding for this work.

**Competing interests:** I have read the journal's policy and I, Kine Johansen, the last authors of this manuscript have the following competing interests to declare: Structured Observation of Motor Performance in Infants (SOMP-I), the assessment methods used in clinical practice at Uppsala University Children's Hospital and tested within the child health services is owned by Barnens rörelsebyrå ekonomisk förening (economic association) Uppsala, Sweden. Kine Johansen is one of the owners of Barnens rörelsebyrå. This does not alter our adherence to PLOS ONE policies on sharing data and materials. The two other authors have no competing interests to disclose.

**Abbreviations:** CHS, child health services; CP, cerebral palsy; GMFCS, gross motor function; classification system; GW, gestational weeks; NICU, neonatal intensive care unit; SOMP-I, structured observation of motor performance in infants.

5 months corrected age, while unilateral spastic CP was most common among those referred by other healthcare professionals or after 5 months corrected age.

## Conclusion

Infants with CP have unequal access to timely physiotherapy, and children considered at low risk for CP receive therapy later. Neonatal follow-up of infants considered at high risk for CP that involves an assessment of motor performance using an evidence-based method during the first months of life corrected age seems to be effective in identifying CP early. Conversely, measuring milestone attainment seems to be a less reliable method for early identification. To provide safe and equal care, all professionals performing developmental surveillance should receive proper training and use evidence-based assessment methods. Physiotherapy should be available prior to formal medical diagnosis.

## Introduction

Cerebral palsy (CP) is a clinical diagnosis used to describe a spectrum of permanent movement disorders caused by non-progressive disturbances that occurred in the developing brain [1,2]. Although the incidence is declining due to improvements in care [2,3], CP is still considered the most common physical disability in childhood, affecting 1.4 per 1000 in high-income countries [3]. Increasing evidence points to the importance of early intervention for motor disorders in children [3–8], and children with milder CP are more responsive to early intervention [8–10]. To intervene during the child's first months of life, to capitalize on the enhanced neuronal plasticity during this critical developmental period, is considered as best practice [3,4,6–8]. Motor training incorporating active, child-initiated, goal-oriented and task-specific movements, as well as family involvement and environmental enrichment, have shown promising results [3,7,8]. Early motor interventions aim to advance motor skills and children's learning potentials as well as to prevent secondary negative consequences of the disorder [3,6,8,11].

In Sweden, children with motor problems are primarily identified through two different healthcare services. The Swedish Child Health Services (CHS), a voluntary free-of-charge primary healthcare service, offer a comprehensive well-child surveillance program to all children aged 0–5 years [12]. During the child's first year of life, the program offers at least 11 scheduled health visits, which include four developmental check-ups and three medical examinations [13]. Motor development is monitored by child healthcare physicians and nurses (specialists in either pediatric care or primary healthcare) observing whether children reach motor milestones or by parental reports [12]. CHS reaches 97% of all Swedish children [14]. Children who have a neonatal clinical history that increases the risk of developmental disorders are additionally enrolled in neonatal follow-up programs guided by national guidelines and adjusted to local routines [15]. At Uppsala University Children's Hospital, these high risk infants are routinely assessed by physiotherapists using an evidence-based assessment method measuring developmental progress and quality of movements at 2, 4 and 10 months corrected age.

Infants with CP or at risk for CP can be accurately diagnosed before 5 months corrected age using clinical reasoning and a combination of standardized tools [6]. However, these high-risk infants only constitute half of all children with CP [5,9,16–18]. Studies have shown that children without readily identifiable risks factors for CP have delayed access to intervention

[16–18]. Additionally, studies have revealed that the Swedish CHS only makes minor contributions to early identification of CP and severe health problems [19,20]. Consequently, children considered at low risk for CP are deprived of interventions that are known to improve outcomes [3,8,9].

The importance of early diagnosis and intervention for children with CP has been understood for decades [2]. Nonetheless, children with CP are still identified late [2,5,16–18], and studies investigating how CP manifests during infancy and what factors affect access to intervention have been called for [17,21,22]. By retrospectively reviewing medical charts, we aimed to describe clinical characteristics of infants with CP as well as what factors enable early access to physiotherapy. We hypothesized that children referred by a neonatologist and those enrolled in neonatal follow-up would have earlier access to physiotherapy and that there would be differences in neonatal clinical history and CP profile compared healthy children born at term. Additionally, because infants can be diagnosed with CP or as being at high risk for CP before 5 months corrected age [6], we chose to investigate differences between children referred for physiotherapy before and after 5 months corrected age to learn more about what enables early access to intervention.

## Methods

### Study population and setting

This retrospective chart review included children born in Uppsala County from 2010 to 2016 who had received a CP diagnosis by July 2019. The children were identified through both the local CP register (n = 38), which is part of the Swedish National Cerebral Palsy Surveillance Program and Registry (CPUP) [23], and the electronic medical chart system (n = 19). As the aim of the present study was to investigate children's access to early physiotherapy, i.e. during the first months of life, children with CP who moved into the county after 1 year of age were excluded from the study (n = 18), leaving 39 eligible children.

In Uppsala County, there are approximately 295,000 inhabitants living in either urban Uppsala or its surrounding rural areas. From 2010 to 2016, 28,691 children were born in the county [24], distributed as follows: 601 born before gestational week (GW) 31 (2.1%), 1517 between GW 32–36 (5.3%) and 26,556 born at term (92.6%). These children attended well-child visits at any of the 44 child health centers in the county. Children who met the defined criteria in the national guideline [15], including all children born before 32 weeks of gestation, were also enrolled in the neonatal follow-up program and had additional visits at Uppsala University Children's Hospital. At 2, 4 and 10 months corrected age, the infants' motor development was routinely assessed by physiotherapists using Structured Observation of Motor Performance in Infants (SOMP-I) [25]. SOMP-I is a non-diagnostic, primarily discriminative assessment method that measures spontaneous and volitional motor performance from term to 12 months of age aiming to identify infants in need of early motor intervention regardless of etiology [25,26]. It allows for a detailed assessment of level of motor development and quality of motor performance in different body parts in different positions. During the study period, regardless of clinical background and the etiology of motor problems, all children aged 0 to 5 years were referred to the same physiotherapy clinic at Uppsala University Children's Hospital for assessment and intervention.

### Data collection

Data collection was conducted during the summer of 2018, with a supplementary search during the summer of 2019. The latter search revealed two additional children who met the inclusion criteria, both born in the fall of 2016. The entries by doctors and physiotherapists

working at the children's hospital were reviewed using a predefined template (S1 Table) based on risk factors mentioned in the review by Novak and colleagues [6]. All data related to age were corrected for prematurity if appropriate (gestational age <37 weeks). The first author performed most of the data collection, while the last author supplemented the data when needed. Any uncertainties were discussed until consensus was reached.

## Measures

*Referral source* was defined as the professional who referred the child to physiotherapy. The children were categorized into two groups: I) referred by a neonatologist or enrolled in neonatal follow-up or II) referred by other healthcare professionals. *Neonatal clinical history* was described using gestational age and number of known risk factors predictive of CP, such as stroke, infections, or seizures, before 5 months corrected age (S1 Table). Gestational age was analyzed using weeks as a continuous variable as well as categorized into extremely preterm (22–27 GW), very preterm (28–32 GW), moderately preterm (32–36 GW) or full-term (37–42 GW). Risk factors were measured as a continuous variable with the exact number of risk factors identified before 5 months corrected age. *Motor severity* was described using the Gross Motor Function Classification System (GMFCS) at 2 years of age [27]. Distinctions between levels are based on functional abilities, where Level I indicates the highest and Level V the lowest functional level. Children at GMFCS Level I-III are ambulatory with or without assistance, while those at GMFCS Level IV-V are considered non-ambulatory and dependent on wheeled mobility. *Type of CP* was described by motor type and topography.

## Statistical analysis

Descriptive statistics were applied to characterize our sample and are presented as number and percentages. Mann-Whitney U tests were used to compare age (months) at first visit to a physiotherapist based on referral source. Linear regression analyses were performed to explore the relationship between age at first visit to a physiotherapist (months) and referral source (categorical). The model was adjusted for number of risk factors (continuous) and severity of motor impairment (categorical). Additionally, we investigated the interaction between referral source and number of risk factors. Gestational age was not included in the model, as preterm infants born before GW 32 are routinely enrolled in the neonatal follow-up and are hence, unless otherwise indicated, assessed by a physiotherapist at 2 months corrected age regardless of the degree of prematurity. The children's neonatal clinical history and CP profile were compared based on referral source and access to physiotherapy before or after 5 months of age using the Mann-Whitney U test, Chi-square analysis or Fischer's exact test. For the analysis of severity of motor impairment, the children were categorized into either ambulatory or non-ambulatory. A two-tailed p-value of <0.05 was considered statistically significant. All calculations were performed using IBM SPSS Statistics 24 for Windows (SPSS Inc., Chicago, IL, USA).

## Ethical approval

Due to the retrospective nature of the study and the risk of losing participants in a small sample, we sought and received approval from the Regional Ethical Review Board in Uppsala, Sweden (reg. no. 2018/173) as well as the Heads of Operations at Uppsala University Children's Hospital and the Habilitation Services in Uppsala to review the data without obtaining informed consent. As data contains potentially identifiable and sensitive patient information data access is limited to by request. Data are available from Uppsala University, Legal Affairs

Division (contact via e-mail: registrator@uu.se), for researchers who meet the criteria for access to confidential data.

## Results

Thirty-nine children met the inclusion criteria, giving a prevalence of CP in Uppsala County of 1.4 cases per 1000 or an incidence of 5.4 cases per year. In one case, local routines had not been followed and this child had delayed access to physiotherapy. Because this case was not representative of the rest of the sample, this participant was excluded from the statistical analysis. There were no missing data.

Two-thirds of the children were born at term (n = 22, 57.9%) (Table 1), and the average gestational age was 38 weeks (min-max 23–41). Slightly more than half of the children were girls (n = 21, 55.3%). Of the 38 children, 22 (58.0%) were referred for physiotherapy by a neonatologist or enrolled in neonatal follow-up (Table 1). The remaining 16 children (42.0%) were referred by other healthcare professionals, i.e., a pediatric neurologist (n = 7) or CHS (n = 9). Twenty-five children (65.8%) were referred before 5 months corrected age.

Twenty-eight children (73.7%) were ambulatory at 2 years of age (Table 1). The largest category of children was assessed at GMFCS Level I (n = 15, 39.5%), while three were assessed at GMFCS Level V (7.9%). The remaining children were equally divided over the GMFCS Levels (II: n = 7 (18.4%), III: n = 6 (15.8%), and IV: n = 7 (18.4%)). The most common types of CP were unilateral (n = 16, 42.1%) and bilateral (n = 17, 44.7%) spastic CP (Table 1). All children with unilateral spastic CP were assessed as ambulatory with or without assistance (GMFCS Level I-III) at 2 years of age. This held true for half of the children with bilateral spastic CP (n = 8, 47.1%). Of the non-ambulatory children (GMFCS Level IV-V), nine (90.0%) were diagnosed with bilateral spastic CP.

### Access to physiotherapy

Children referred by a neonatologist or enrolled in neonatal follow-up had significantly earlier access to physiotherapy compared to children referred by other healthcare professionals (p < 0.0001) (Table 1). On average, these infants were approximately two months old (min-max: -1-4) at the first visit to the physiotherapist (Table 1). The corresponding age for infants referred by other healthcare professionals was approximately 7.5 months (min-max 1–24). All children referred by a neonatologist or enrolled in neonatal follow-up had their first visit before 5 months corrected age (n = 22, 100%), while this was only true for three children (18.8%) referred by other healthcare professionals. The unadjusted regression model demonstrated that both referral source and number of risk factors were associated with early access to physiotherapy (Table 2). When the variables were entered into the adjusted regression model, referral source explained unique variance in predicting age at first visit to a physiotherapist ($R^2$ 0.550, B 4.213, p < 0.0001). There was an interaction between referral source and the number of risk factors (Table 2), and this was especially true for the children referred from other healthcare professionals (Fig 1).

### Comparisons of neonatal clinical history and CP profile

**Referral source.** Children referred by a neonatologist or enrolled in neonatal follow-up were born significantly more prematurely than children referred by other healthcare professionals (p = 0.004) (Table 1). They had more known risk factors predictive of CP (p <0.0001) (Table 1), with a mean (SD) of 5.6 (2.5) compared to 0.9 (1.5) in children referred by other healthcare professionals. No statistically significant difference was found in severity of motor impairment based on referral source (Table 1). However, more children referred by a

**Table 1. Descriptive statistics for all children with cerebral palsy (CP) as well as divided by referral source and referred before or after 5 months corrected age.**

| | | | Referral source | | | Before or after 5 months | | |
|---|---|---|---|---|---|---|---|---|
| | | **Total** | **Neo[a]** | **Other[b]** | **P-value** | **0–4 mo[c]** | **5–24 mo[c]** | **P-value** |
| | | **n = 38** | **n = 22 (58)** | **n = 16 (42)** | | **n = 25 (66)** | **n = 13 (34)** | |
| Sex | | | | | | | | |
| | Boys, n (%) | 17 (44.7) | 5 (22.7) | 12 (75.0) | | 8 (32.0) | 9 (69.2) | |
| | Girls, n (%) | 21 (55.3) | 17 (77.3) | 4 (25.0) | | 17 (68.0) | 4 (30.8) | |
| Birth history | | | | | | | | |
| | **Gestational age, weeks** | | | | | | | |
| | Mean (SD) | 35.2 (5.5) | 32.6 (5.8) | 38.6 (2.1) | 0.004 | 33.3 (5.8) | 38.7 (2.2) | 0.009 |
| | Median | 38 | 30.5 | 39 | | 33 | 39 | |
| | Min-max | 23–41 | 23–41 | 33–41 | | 23–41 | 33–41 | |
| | **Gestational age groups** | | | | | | | |
| | 22–27 GW[d] | 4 (10.5) | 4 (18.2) | 0 (0) | 0.005 | 4 (16.0) | 0 (0) | 0.015 |
| | 28–31 GW[d] | 8 (21.1) | 8 (36.4) | 0 (0) | | 8 (32.0) | 0 (0) | |
| | 32–36 GW[d] | 4 (10.5) | 2 (9.1) | 2 (12.5) | | 3 (12.0) | 1 (7.7) | |
| | 37–42 GW[d] | 22 (57.9) | 8 (36.4) | 14 (87.5) | | 10 (40.0) | 12 (92.3) | |
| | **Known risk factors** | | | | | | | |
| | Mean (SD) | 3.6 (3.1) | 5.6 (2.5) | 0.9 (1.5) | < 0.0001 | 5.3 (2.5) | 0.3 (0.5) | < 0.0001 |
| | Median | 4 | 6 | 0 | | 6 | 0 | |
| | Min-max | 0–9 | 0–9 | 0–5 | | 0–9 | 0–1 | |
| CP profile | | | | | | | | |
| | **Severity of motor impairment** | | | | | | | |
| | Ambulatory | 28 (73.7) | 14 (63.6) | 14 (87.5) | 0.143 | 16 (64.0) | 12 (92.3) | 0.118 |
| | Non-ambulatory | 10 (26.3) | 8 (36.4) | 2 (12.5) | | 9 (36.0) | 1 (7.7) | |
| | **Type of CP** | | | | | | | |
| | Unilateral | 16 (42.1) | 5 (22.7) | 11 (68.8) | 0.012 | 7 (28.0) | 9 (69.2) | 0.026 |
| | Bilateral | 17 (44.7) | 14 (63.6) | 3 (18.8) | | 15 (60.0) | 2 (15.4) | |
| | Ataxic/dyskinetic | 5 (13.2) | 3 (13.6) | 2 (12.5) | | 3 (12.0) | 2 (15.4) | |
| Physiotherapy | | | | | | | | |
| | **First visit, months** | | | | | | | |
| | Mean ±SD | 4.6 (5.3) | 1.5 (1.2) | 8.8 (5.8) | < 0.0001 | 1.6 (1.3) | 10.2 (5.5) | < 0.0001 |
| | Median | 2 | 1.9 | 7.6 | | 2 | 8.6 | |
| | Min-max | -1–24 | -1–4 | 1–24 mo | | -1–4 | 5–24 mo | |
| | **Referred before 5 months** | | | | | | | |
| | Yes | 25 (65.8) | 22 (100) | 3 (18.8) | < 0.0001 | | | |
| | No | 13 (34.2) | 0 (0) | 13 (81.3) | | | | |

Results are presented in numbers and percentage. Mean and standard deviation (SD) plus median, minimum and maximum are presented when applicable. Mann-Whitney U test, Chi-square analysis or Fischer's exact test are used to compare groups.

[a]Neo: Neonatologists or enrolled in neonatal follow-up.

[b]Other: Other healthcare professionals.

[c]Mo: Months.

[d]GW: Gestational weeks.

neonatologist or enrolled in the neonatal follow-up were assessed as non-ambulatory (n = 8, 80.0%) compared to those referred by other healthcare professionals (n = 2, 20.0%). Children assessed as ambulatory with or without assistance were divided equally between the referral sources (Table 1). There was a statistically significant difference between the groups regarding type of CP (p = 0.012) (Table 1), such that the children referred by a neonatologist or enrolled

**Table 2. Linear regression models.**

| Independent variables | R² | B | 95% CI | P |
|---|---|---|---|---|
| **Unadjusted** | | | | |
| Referral source (other[a] vs neo[b]) | 0.477 | -7.291 | (-9.88, -4.71) | < 0.0001 |
| Number of risk factors | 0.465 | -1.151 | (-1.58, -0.73) | < 0.0001 |
| Severity of motor problem (GMFCS[c] Levels I-V) | 0.152 | -1.489 | (-2.68, -0.30) | 0.016 |
| Interaction referral source * number of risk factors | 0.378 | -0.973 | (-1.40, -0.55) | < 0.0001 |
| **Adjusted** | | | | |
| *Model 1* | 0.550 | | | < 0.0001 |
| Referral source (other[a] vs neo[b]) | | -4.213 | (-7.93, -0.50) | 0.027 |
| Number of risk factors | | -0.571 | (-1.18, 0.04) | 0.066 |
| Severity of motor problem (GMFCS[c] Levels I-V) | | -0.429 | (-1.41, 0.55) | 0.381 |
| *Model 2* | 0.664 | | | < 0.0001 |
| Referral source (other[a] vs neo[b]) | | -8.085 | (-12.11, -4.06) | < 0.0001 |
| Number of risk factors | | -2.218 | (-3.36, -1.08) | < 0.0001 |
| Severity of motor problem (GMFCS[c] Levels I-V) | | -0.553 | (-1.42, 0.31) | 0.202 |
| Interaction referral source * number of risk factors | | 2.107 | (0.82, 3.39) | 0.002 |

Summary of unadjusted and adjusted linear regressions models for variables hypothesized to predict access to early physiotherapy (months corrected age), where R² denotes explained variance in the model and B the degree of change in the outcome variable (months corrected age) for every 1-unit of change in the dependent variable.

[a]Other: Other healthcare professionals.

[b]Neo: Neonatologists or enrolled in neonatal follow-up.

[c]GMFCS: Gross Motor Function Classification System.

in the neonatal follow-up were more often diagnosed with bilateral spastic CP (n = 14, 63.6%) and those referred by other healthcare professionals with unilateral spastic CP (n = 11, 68.8%).

**Referred before and after 5 months corrected age.** When analyzing differences in neonatal clinical history and CP profile in relation to whether the child had access to physiotherapy before or after 5 months of age, similar results were found. Children referred to physiotherapy before 5 months corrected age were significantly more often born prematurely

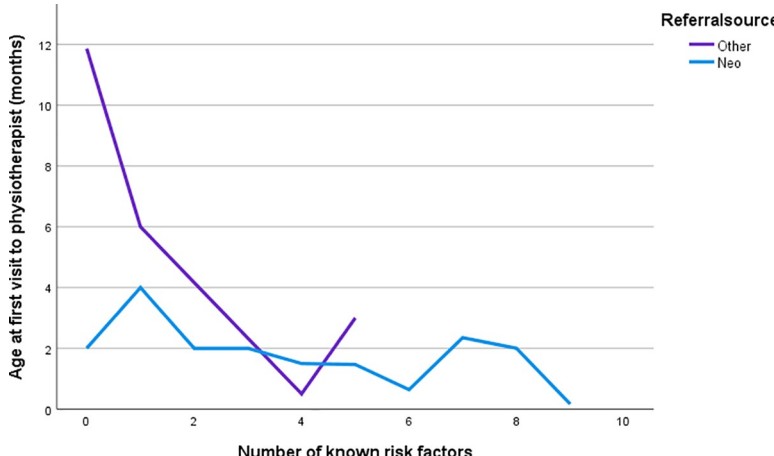

**Fig 1. The interaction between referral source and number of risk factors.** The total number of known risk factors before 5 months corrected age (X-axis) and the mean corrected age at first visit to physiotherapist in months (Y-axis). The lines represent the referral source.

(p = 0.009) and had more known risk factors predictive of CP (p < 0.0001) (Table 1). Most children (n = 9, 69.2%) referred after 5 months of age had no known risk factors predictive of CP (Table 1), while the corresponding figure (mean (SD)) for those referred before 5 months corrected age was 5.3 (2.5). For the children referred by other healthcare professionals, the number of risk factors was vital for early access to physiotherapy (Fig 1). Additionally, children referred after 5 months corrected age had milder forms of CP, where all, except one, were ambulatory with or without assistance at 2 years of age (n = 12, 92.3%) (Table 1). No statistically significant difference was found in severity of motor impairment when comparing the two groups. However, the majority (n = 9, 90.0%) of the children assessed as non-ambulatory had their first visit to physiotherapists before 5 months of age. Regarding type of CP, there was a statistically significant difference between the two groups (p = 0.026) (Table 1), such that those referred before 5 months corrected age were more often diagnosed with bilateral spastic CP (n = 15, 60.0%). Among those referred after 5 months corrected age, unilateral spastic CP was most common (n = 9, 69.2%).

## Discussion

Our study confirmed our hypothesis that children referred by a neonatologist and those enrolled in neonatal follow-up had earlier access to physiotherapy compared to children referred by other healthcare professionals, even when the number of known risk factors and the severity of motor impairment were controlled for. However, the number of risk factors seemed to be vital for early access to physiotherapy for the children that were not NICU graduates. Furthermore, the groups differed on all variables in regards in neonatal clinical history and CP profile. The prevalence of CP was similar to figures reported in other high-income countries [3], and only a little more than half of the children were referred by a neonatologist or enrolled in neonatal follow-up. Children born at term constituted 58% of all CP cases. Among all the children, 66% had access to physiotherapy before 5 months corrected age.

Similar to Boychuck and colleagues' [17] findings, referral source and number of known risk factors predicted earlier access to physiotherapy. It is important to acknowledge that children who require neonatal intensive care are considered to be at high risk for CP and are routinely enrolled in neonatal follow-up programs [6]. In this setting, professionals know they are assessing high-risk children. The children may also have more severe impairments, which may partly explain why CP in these children are identified earlier [17]. Additionally, in neonatal follow-up, children are assessed by physiotherapists using evidence-based assessment methods [15]. Being trained and skilled in assessing early motor performance using evidence-based assessment methods seemed to improve early identification of children with CP.

However, only slightly more than half of all children with CP are enrolled in neonatal follow-up [5,9,16–18]. For the remaining children, age at referral varies considerably [16–18]. In this connection, the primary healthcare services, including CHS, show the greatest delays [16–19]. Although the Swedish CHS is striving to improve their methods and the infants in our study were identified at an earlier age than in previous studies [16–19], the children were nevertheless referred later than high-risk infants and with greater variability. In our study, the children who were not NICU graduates had few, if any, known risk factors for CP, but having more risk factors led to earlier access to physiotherapy. Furthermore, the majority of all children were assessed as ambulatory at 2 years of age. Not having identifiable risk factors for CP as well as having a high functional level often leads to delayed referral for intervention [5,9,18,28]. Adding this to the relatively low prevalence of CP indicates that very few healthcare professionals will ever be the first assessor of an infant with emerging CP. This highlights the challenge of early identification of children with CP outside neonatal follow-up. Thus, our

question should be: How can we enable early identification of children with CP without known risk factors for CP? In Sweden, healthcare professionals working within CHS could play a pivotal role, as nearly all children pass through this service. However, to enable this, we need to learn more about early signs of CP and its development during the first year of life.

Additionally, we need to consider how motor development is assessed within CHS. We know that, when our aim is early identification of children with CP, assessments focusing on motor milestones are not reliable [9]. Some children with CP may have near normal milestone attainment over the first year of life [2,9], and this is especially true for children with mild unilateral CP [9]. Merely assessing how far the child has come in its development can lead to a false impression of typical development, and this is why assessing quality of movement is recommended [7,9,16]. Children with unilateral or bilateral spastic CP, especially those with a milder motor impairment, are usually identified late [18,28]. Being observant of early quality deviations such as persistent asymmetries, especially early hand preference [6,9,10,22], irrespective of motor delay, should warrant a more thorough examination and prompt referral to physiotherapy.

To enable timely identification of children with CP, evidence-based assessment methods should be an integrated part of developmental surveillance. Standardizing care could potentially reduce variability across organizations and professionals [29], hence increasing equality and patient safety. Furthermore, using such methods improves accuracy, enables earlier identification (including mild delays or suspected deviations) and provides more information compared to clinical judgement alone [6,9,30,31]. Not using such methods will delay identification of children with CP [16–18], consequently depriving children of interventions known to be beneficial [3,4,6,8]. The fact that the rate of CP is falling and that CP severity is lessening further highlight this need [3]. A recent scoping review suggested that feasible methods for well-child surveillance are lacking [22]. Johansen and colleagues [32], however, showed that when child health nurses used SOMP-I in routine well-child surveillance, the method appeared to be clinically useful. Providing child healthcare professionals with an evidence-based assessment method may enable earlier identification of children with CP.

All professionals performing developmental surveillance should receive adequate training, use evidence-based assessment methods when available and be skilled in discriminating atypical movement from variations in typical movement [6]. However, it is important to remember that proper assessment of infants using any standardized method is an acquired skill requiring practice over time [29]. Nurses learning to use SOMP-I stressed that becoming a proficient assessor requires training and practice [33]. Furthermore, infants displaying aberrant motor performance should have access to physiotherapy prior to any formal medical diagnosis, as mild cases without any risk factors can be particularly difficult to diagnose [9]. Important to note is that the linear serial model of referral should be avoided, as it prolongs lead time and delays intervention [16,17].

## Limitations

Our results are based on a small clinical population from one Swedish county. Nevertheless, the prevalence of CP was similar to figures reported elsewhere, indicating that it is a representative sample. However, a larger sample could have revealed statistically significant differences between severity of the motor problems and referral source or access to physiotherapy before or after 5 months corrected age. Furthermore, the youngest children were only 2 years of age when we conducted the initial search, which entails the risk that not all children with CP were identified. However, no child was identified after 2 years of age, and only two children meeting the inclusion criteria were identified in the supplementary search. The retrospective design

limited our access to the entries made at the children's hospital, possibly resulting in missed information from other sources, such as CHS. Furthermore, the review of the medical charts was mainly performed by the first author. To reduce the risk of bias, a template was used when collecting the data (S1 Table).

## Clinical implications

Early identification and timely intervention are widely accepted as best practice for children with CP. Our study indicates that a neonatal follow-up program that includes infants at high risk for CP as well as those born before 32 weeks of gestation and that involves an assessments of motor performance using an evidence-based method during the first months of life corrected age is effective in identifying children with CP early. However, infants have unequal access to timely physiotherapy, and children considered at low risk for CP receive therapy later. The high variability in referral practices suggests that knowledge of CP characteristics in infants varies widely across organizations and professionals. It is especially children without readily identifiable risk factors who have delayed access to intervention, indicating that the current CHS practice of measuring milestone attainment does not effectively identify CP. This is concerning, given CHS's key role in early identification of developmental disorders.

To provide high quality care that is equal and safe, all professionals working with developmental surveillance, regardless of organizational level, need appropriate knowledge of typical and atypical motor development, including early clinical signs of CP. Without continuous professional development initiatives to increase such knowledge, children will continue to experience delayed access to physiotherapy. Evidence-based assessment methods should be used when available, as these improve early identification and support clinical decision-making. Furthermore, when aberrant motor performance is observed, the child should be promptly referred to physiotherapy, regardless of formal medical diagnosis.

## Supporting information

**S1 Table. Template for data collection.**
(DOCX)

## Acknowledgments

We wish to thank Kristina Persson (Associate Professor and physiotherapist) for her valuable feedback during preparation of the manuscript, as well as to acknowledge the collaboration provided by the Habilitation Services in Uppsala County, Sweden.

## Author Contributions

**Conceptualization:** Linnéa Hekne, Cecilia Montgomery, Kine Johansen.

**Data curation:** Kine Johansen.

**Formal analysis:** Linnéa Hekne, Cecilia Montgomery, Kine Johansen.

**Funding acquisition:** Kine Johansen.

**Investigation:** Linnéa Hekne, Kine Johansen.

**Methodology:** Linnéa Hekne, Cecilia Montgomery, Kine Johansen.

**Project administration:** Kine Johansen.

**Resources:** Kine Johansen.

**Software:** Kine Johansen.

**Supervision:** Kine Johansen.

**Visualization:** Kine Johansen.

**Writing – original draft:** Linnéa Hekne, Kine Johansen.

**Writing – review & editing:** Cecilia Montgomery, Kine Johansen.

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
