## [Decision Letter · Decision Letter 0]

15 Feb 2021

PONE-D-20-28394

Unequal access to timely physiotherapy for children with Cerebral Palsy: a retrospective chart review

PLOS ONE

Dear Dr. Johansen,

Thank you for submitting your manuscript to PLOS ONE. After careful consideration, we feel that it has merit but does not fully meet PLOS ONE’s publication criteria as it currently stands. Therefore, we invite you to submit a revised version of the manuscript that addresses all points raised during the review process.

We look forward to receiving your revised manuscript.

Kind regards,

Christos Papadelis, Ph.D.

Academic Editor

PLOS ONE

Journal Requirements:

2. In your Methods section, please provide additional information regarding the participant inclusion and exclusion criteria. Furthermore,  we note that you have reported significance probabilities of 0 in places. Since p=0 is not strictly possible, please correct this to a more appropriate limit, eg 'p<0.0001'

"SOMP-I is owned by Barnens rörelsebyrå ekonomisk förening (economic association) Uppsala, Sweden. Kine Johansen is a partner of Barnens rörelsebyrå. All other authors have no conflicts of interest to disclose."

Reviewers' comments:

Reviewer's Responses to Questions

**Comments to the Author**

1. Is the manuscript technically sound, and do the data support the conclusions?

Reviewer #1: Partly

Reviewer #2: Partly

2. Has the statistical analysis been performed appropriately and rigorously? 

Reviewer #1: Yes

Reviewer #2: Yes

3. Have the authors made all data underlying the findings in their manuscript fully available?

Reviewer #1: No

Reviewer #2: No

4. Is the manuscript presented in an intelligible fashion and written in standard English?

Reviewer #1: Yes

Reviewer #2: Yes

5. Review Comments to the Author

Reviewer #1: The authors retrospectively reviewed patient data from two health care systems in a county in Sweden in the aim to assess whether children with cerebral palsy (CP) have equal access to timely physiotherapy. They concluded from the data that Children with CP have unequal access to physiotherapy. The authors emphasized the importance of timely access of physiotherapy to children with CP. The data and results might be helpful for the improvement of local health policy for children with CP.

The authors provided detailed descriptive results of data. However, the description of data collection is not detailed. Several aspects of information as the following should be described:

(1) as a retrospective review, do you have a data collection guidance or standard, which is like on what data to be reviewed , how to collect data, and data quality control, before the formal data review? if so, please provide a data collection flow chart.

(2) as described in the manuscript, the first author reviewed most of the data from two different health care systems, is there the possibility of any potential bias on reviewing data from two different health care systems mostly by one researcher, has any potential bias been considered to be controlled before formal data review, has any preventive method been used to control the bias?

(3) the authors did not mention any data missing in the manuscript. How has data missing been handled if you did have the cases of missing data?

As to statistical analysis, the authors used Kruskal-Wallis H Test but without providing any justification for the selection of this method.

As to results, the authors found that both referral source and number of risk factors have significant influences to the first visit to physiotherapy (see Table 2), and concluded that “children referred from the child health services have the most delayed access” (see abstract conclusion). A likely scenario is that number of risk factors dominates the decision of the first visit to physiotherapy but children with CP from the child health services have the fewer number of risk factors compared to other referral sources, and the fewer number of risk factors caused the delay of referral of children with CP to physiotherapy by the child health services. If the number of risk factors was controlled, is the conclusion “children referred from the child health services have the most delayed access” still correct?

Reviewer #2: The purpose of the review is to provide the editors with an expert opinion regarding the validity and quality of the manuscript under consideration. The review should also supply authors with explicit feedback on how to improve their papers so that they will be acceptable for publication in PLOS ONE. As you write, consider the following points:

• What are the main claims of the paper and how significant are they for the discipline?

I appreciate the opportunity to review the manuscript named “Unequal access to timely physiotherapy for children with Cerebral Palsy: a retrospective chart review”

-The authors claims to investigate whether children with cerebral palsy (CP) in Sweden have equal access to timely physiotherapy. Additionally, the children’s birth history and CP profile are described to understand typical features that might enable earlier identification for children with CP in the future. Equity in care is important and it is clearly of interest to know if it is provided for children with CP, as that would be a prerequisite in order to correctly target suitable measures. However, I would also suggest nuancing the results with regard to motor function and need of physiotherapy, further discussed in comments below.

• Are the claims properly placed in the context of the previous literature? Have the authors treated the literature fairly?

-Yes, however the literature may also provide further information for nuancing according to gross motor function, mentioned above.

• Do the data and analyses fully support the claims? If not, what other evidence is required?

-Data sufficiently supports the claims. Multilevel regressions may be considered.

• PLOS ONE encourages authors to publish detailed protocols and algorithms as supporting information online. Do any particular methods used in the manuscript warrant such treatment? If a protocol is already provided, for example for a randomized controlled trial, are there any important deviations from it? If so, have the authors explained adequately why the deviations occurred?

-No need.

• If the paper is considered unsuitable for publication in its present form, does the study itself show sufficient potential that the authors should be encouraged to resubmit a revised version?

-Yes, I suggest a revised version to be resubmitted

• Are original data deposited in appropriate repositories and accession/version numbers provided for genes, proteins, mutants, diseases, etc.?

-Not applicable

• Are details of the methodology sufficient to allow the experiments to be reproduced?

-Yes

• Is the manuscript well organized and written clearly enough to be accessible to non-specialists?

-The manuscript should be revised more thoroughly for structure and readability, suggestions follows below.

Although confidential comments to the editors are respected, any remarks that might help to strengthen the paper should be directed to the authors themselves.

Explicit feedback for improvement:

I thank you for letting me read and comment on your work, which I find important and interesting. I start of with general comments and proceed with specific comments referring to line numbers.

I would recommend you to ethically reconsider some of the presented data due to confidentiality of the participants, further commented below.

I would think it is of interest to nuance the discussion regarding need for physiotherapy and motor function. This study, like earlier cited studies shows roughly that when disability is not expected or fairly invisible, diagnosis/referral is more likely to be delayed. Conceivably these children are not the ones most in need of physiotherapy, thus, the consequences of less early physiotherapy may not be quite generalizable for all children with CP. For ease of reading, consider using the same wording throughout the document, e.g. medical history/complicated birth history, and also to provide both numbers and percentages.

Introduction:

51: Please provide a reference for this statement.

52-53: Children with complicated birth history are enrolled to neonatal follow-up, which may be confused with information in methods, lines 79-80 where neonatal follow-up is only for children born <32 weeks. Please clarify criteria for neonatal follow-up

54-56: I would appreciate a more specific description of the knowledge gap – why this needs to be explored again but specifically for physiotherapy.

57: “Improve outcomes” is very general, please specify/exemplify interventions and outcomes and possibly for which children (GMFCS)

63: Is it possible that early motor interventions are more important for children with GMFCS III-V, and thereby lessening the impact of the inequality in timely physiotherapy, as all but one(!) from the well child visits had GMFCS I-II? While such information should not lead to disregarding of inequality, it would all the same nuance the picture.

Method:

72-74: Exclusion criteria seems to exclude all children not born in Uppsala regardless of when they moved to Uppsala? Please clarify the exclusion criteria and the reasons for it as excluded children correspond to almost 50% of your current sample

78-79: Do all term born children always participate in the well child program?

79-80: Please clarify which children receives neonatal follow-up and how neonatologist differs from neuropediatrician. According to line 52-53, neonatal follow-up would not only include <32 weeks. Do the preterm children also participate in the well child program? 82: The neuropediatricians seem to be part of the neonatal follow-up, which further complicates the relevance of this division of referral source, see lines 52 and 79.

102: Please consider if multilevel analysis would be useful, e.g. multivariable logistic regressions.

109, Ethics: I disagree with your statement that all data are presented on group level. Some groups consists of single children. In a fairly small community of children with CP and their parents, in a named region in Sweden, there would be a significant risk that individual children are identified by some of these data. This would be particularly unfortunate since no informed consent is collected.

Results:

117-118: Please clarify the meaning of “missed early” and why this is reason for exclusion.

Table 1: In the rows you explain that you report n (%), however the percentages are summed up by column. I would recommend you to reconsider this way of presenting in order to gain readability

120-121: Please explain the relevance of the difference between being referred by neonatologist or neuropetiatrician – are these children not part of the same neonatal follow-up? Also, is it possible that CHS does not refer to physiotherapist directly, but via e.g. neuropediatricians, thus contributing to the delay?

Table 2: It would be helpful if you explain H and Df among the abbreviations in the table and also elaborate the table heading so that the table can be understood separately from the text. GMFCS is quite close to significant regarding first visit to physiotherapy – I propose you add to the discussion whether your results actually show that there is no difference with regard to GMFCS or if the results may be due to power-issues.

Discussion:

169: It is a bit unclear if you refer to the Ref 12 or to own results (or both)

172: As mentioned above, I find it a difficult to follow this grouping of children due to referral source – which are the children who are not from CHS and not in the neonatal follow-up There is a lack of consistency and/or a need of a more concise description throughout the manuscript

175-176: Again, as mentioned earlier, I would suggest you elaborate around this matter. While it is important to provide equal care, as to receive early FT assessment, there may also be limited consequences for children with mild motor difficulties to be referred to physiotherapy at a higher age, because of their fairly good motor function(?). Other difficulties associated with CP may be prioritized when motor function is good, such as e.g. speech or cognition.

178: Please clarify if the early interventions refer to physiotherapy or other interventions. Please consider discussion about power in your study when discussing GMFCS as not significantly associated with early referral to physiotherapist.

205-206: It seems in this statement like all children are not included in the neonatal follow-up – is that right and if so, please state this more clearly. According to the introduction line 52 all children with complicated birth history are routinely included.

206: Please specify which interventions and which guidelines you refer to.

216: Please specify the kind of review you refer to.

220-232: To implement use of evidence based motor assessment tools within CHS seems to be a suitable suggestion in order to early detect more children with presumed CP. I suggest you emphasize this even more. (As there is a shortage of specialist CHS nurses in primary care, maybe such assessment could be introduced as a cooperation task for primary health care physiotherapists in order to widen their area of responsibility and thereby strengthening their professional role?)

6. PLOS authors have the option to publish the peer review history of their article (what does this mean?). If published, this will include your full peer review and any attached files.

Reviewer #1: **Yes: **Yanlong Song

Reviewer #2: No

---

## [Author Response · Author response to Decision Letter 0]

31 Mar 2021

Response to reviewers comments regarding manuscript PONE-D-20-28394

We would like to start by thanking the academic editor and the reviewers for their valuable and constructive feedback! Below you find our response to their recommended and suggested revisions. Based on the feedback from the reviewers the revisions are extensive. We have refined our aim and research questions to be more concise, and as a result we needed to redo the statistical analysis as well as edit tables and exclude the figure. To address these changes as well as to meet the requests by the reviewers we have rewritten parts of the introduction and discussion as well. 

We have highlighted the changes in the manuscript in yellow. Given the extensive revision of the manuscript it has also been edited for language. These language changes are not highlighted to improve the readability of the text. If necessary, we can provide the manuscript with these tracked changes. Added text is highlighted in yellow, while removed text appears as strikethrough and also highlighted in yellow.

Journal Requirements

We have edited the manuscript to meet PLOS ONE’s style requirements. 

(2) In your Methods section, please provide additional information regarding the participant inclusion and exclusion criteria. Furthermore, we note that you have reported significance probabilities of 0 in places. Since p=0 is not strictly possible, please correct this to a more appropriate limit, eg 'p<0.0001'

We have edited this throughout the manuscript. 

(3) We note that the grant information you provided in the ‘Funding Information’ and ‘Financial Disclosure’ sections do not match. When you resubmit, please ensure that you provide the correct grant numbers for the awards you received for your study in the ‘Funding Information’ section.

As none of the authors have received any specific funding for this work, we have removed the previous reported funding bodies and instead ticked the box for ‘The author(s) received no specific funding for this work.’ Could you please update the financial disclosure in the online submission form to: “The authors received no specific funding for this work”? Thank you in advance! 

(4) Thank you for stating the following in the Competing Interests section: "SOMP-I is owned by Barnens rörelsebyrå ekonomisk förening (economic association) Uppsala, Sweden. Kine Johansen is a partner of Barnens rörelsebyrå. All other authors have no conflicts of interest to disclose." Please confirm that this does not alter your adherence to all PLOS ONE policies on sharing data and materials, by including the following statement: "This does not alter our adherence to PLOS ONE policies on sharing data and materials.” 

We have updated the statement regarding competing interests to adhere to PLOS ONE policies and guidelines, see below. The statement is included in the cover letter. Please change the online submission form on our behalf. 

I have read the journal's policy and I, Kine Johansen, the last authors of this manuscript have the following competing interests to declare: Structured Observation of Motor Performance in Infants (SOMP-I), the assessment methods used in clinical practice at Uppsala University Children’s Hospital and tested within the child health services is owned by Barnens rörelsebyrå ekonomisk förening (economic association) Uppsala, Sweden. Kine Johansen is one of the owners of Barnens rörelsebyrå. This does not alter our adherence to PLOS ONE policies on sharing data and materials. The two other authors have no competing interests to disclose.

(5) We note that you have indicated that data from this study are available upon request. PLOS only allows data to be available upon request if there are legal or ethical restrictions on sharing data publicly. 

We have updated the data availability statement to adhere to PLOS guidelines, see below. The statement is included in the cover letter. Please change the online submission form on our behalf. We have also added this information to the section ‘Ethical approval’ in the manuscript. 

Data cannot be shared publicly because it includes potentially identifying and sensitive patient information, therefore access is limited to by request by the Regional Ethical Review Board in Uppsala, Sweden (reg. no. 2018/173). Data are available from Uppsala University, Department of Women’s and Children’s Health (contact via e-mail: kine.johansen@kbh.uu.se) for researchers who meet the criteria for access to confidential data.

Review comments to the Author

Reviewer #1

The authors provided detailed descriptive results of data. However, the description of data collection is not detailed. Several aspects of information as the following should be described:

(1) As a retrospective review, do you have a data collection guidance or standard, which is like on what data to be reviewed, how to collect data, and data quality control, before the formal data review? if so, please provide a data collection flow chart.

Besides the inclusion and exclusion criteria and the data collection template (S1 Table), we did not have a data collection guidance on standard. Hence, we cannot provide a data collection flow chart. 

(2) As described in the manuscript, the first author reviewed most of the data from two different health care systems, is there the possibility of any potential bias on reviewing data from two different health care systems mostly by one researcher, has any potential bias been considered to be controlled before formal data review, has any preventive method been used to control the bias?

All data collected and used in this study is based on data from hospital stays and visits to the Uppsala University Children’s Hospital. All children with motor problems were during the study period referred for to the same pediatric physiotherapy clinic located at the children’s hospital. CP type and typography as well as the Gross Motor Function Classification System (GMFCS) level was collected from the assessment by the pediatric neurologist around 2 years of age. No data were collected from the child health services. 

(3) The authors did not mention any data missing in the manuscript. How has data missing been handled if you did have the cases of missing data?

As all data are based on medical records and we only used data available from the records we did not have any missing data. We have added a sentence in the first section of the results section. 

(4) As to statistical analysis, the authors used Kruskal-Wallis H Test but without providing any justification for the selection of this method. As to results, the authors found that both referral source and number of risk factors have significant influences to the first visit to physiotherapy (see Table 2), and concluded that “children referred from the child health services have the most delayed access” (see abstract conclusion). A likely scenario is that number of risk factors dominates the decision of the first visit to physiotherapy but children with CP from the child health services have the fewer number of risk factors compared to other referral sources, and the fewer number of risk factors caused the delay of referral of children with CP to physiotherapy by the child health services. If the number of risk factors was controlled, is the conclusion “children referred from the child health services have the most delayed access” still correct? The number of risk factors plays a considerable part in referral to physiotherapy. However, as the separate variables are dependent on each other, i.e. if a child have more risk factors they are more likely to be admitted to the NICU and hence be included in the neonatal follow-up, we cannot do this kind of analysis. 

We agree with reviewer 1 that referral source and the number of risk factors are dependent on each other. As reviewer propose children with more risk factors were admitted to the neonatal intensive care unit and had earlier access to physiotherapy. To analyze this relationship we performed a multivariate regression analyses controlling for the number of risk factors and severity of the motor problem. We found that referral source contributed with unique variance compared to the number of risk factors. Our interpretation of these findings are that our current system of identifying CP in high risk infants is effective, i.e. a generous neonatal follow-up program which includes children with known risk factors predictive of CP as well as those born before 32 weeks of gestation and that involves an early assessment of motor performance. After redefining our aim and research questions we removed the Kruskal-Wallis H Test from our analysis. 

Reviewer #2

• Are the claims properly placed in the context of the previous literature? Have the authors treated the literature fairly? Yes, however the literature may also provide further information for nuancing according to gross motor function, mentioned above.

We have refined the aim and research questions, and rewritten the introduction, results and discussion. Hopefully this has provided a more nuanced picture of motor function and the need of timely motor interventions. The effects of early motor interventions holds for both gross and fine motor function. 

• Do the data and analyses fully support the claims? If not, what other evidence is required? Data sufficiently supports the claims. Multilevel regressions may be considered. 

Thank you for the suggestion. We have redone our statistical analysis and included a univariate and multivariate regression analyses. 

• Is the manuscript well organized and written clearly enough to be accessible to non-specialists? The manuscript should be revised more thoroughly for structure and readability, suggestions follows below.

Response to explicit feedback for improvement: 

- I would recommend you to ethically reconsider some of the presented data due to confidentiality of the participants, further commented below.

We have address this in the comments below.

- I would think it is of interest to nuance the discussion regarding need for physiotherapy and motor function. This study, like earlier cited studies shows roughly that when disability is not expected or fairly invisible, diagnosis/referral is more likely to be delayed. Conceivably these children are not the ones most in need of physiotherapy, thus, the consequences of less early physiotherapy may not be quite generalizable for all children with CP. 

We have address this in the comments below.

- For ease of reading, consider using the same wording throughout the document, e.g. medical history/complicated birth history, and also to provide both numbers and percentages.

We have changed medical history/complicated birth history to neonatal clinical history as well as added both number and percentages throughout the manuscript.

Introduction

• 51: Please provide a reference for this statement.

We have added a reference as suggested. 

• 52-53: Children with complicated birth history are enrolled to neonatal follow-up, which may be confused with information in methods, lines 79-80 where neonatal follow-up is only for children born <32 weeks. Please clarify criteria for neonatal follow-up. 

We have rewritten the ‘Introduction’ as well as the ‘Study population and setting’ to clarify the inclusions criteria for neonatal follow-up in Uppsala County. 

• 54-56: I would appreciate a more specific description of the knowledge gap – why this needs to be explored again but specifically for physiotherapy.

We have refined the aim and research question and highlighted the knowledge gap in the last section of the introduction. 

• 57: “Improve outcomes” is very general, please specify/exemplify interventions and outcomes and possibly for which children (GMFCS)

Early intervention is in this manuscript primarily referring to early motor interventions. We have added two sentences elaborating on this in the first section of the introduction.

According to Novak et al (2020) the following features are common for effective motor interventions for children with CP: ‘practice of real-life tasks and activities, using self-generated active movements, at a high intensity, where the practice directly targets the achievement of a goal set by the child (or a parent proxy if necessary). The mechanism of action is experience-dependent plasticity. Motivation and attention are vital modulators of neuroplasticity, and successful task-specific practice is rewarding and enjoyable to children, producing spontaneously regular practice.’ 

Given that children with less severe motor impairment are more likely to be able to produce active goal-directed movements, they are also those who are more responsive to early motor interventions (Morgan et al., 2018; Sakzewski et al., 2019). Early motor interventions aim to improve motor outcomes as well as to prevent secondary negative consequences of not being able to move age-appropriately.

Early motor intervention programs have been associated with beneficial effects of both motor and cognitive outcomes (Hadders-Algra, 2021; Morgan et al., 2016).

• 63: Is it possible that early motor interventions are more important for children with GMFCS III-V, and thereby lessening the impact of the inequality in timely physiotherapy, as all but one(!) from the well child visits had GMFCS I-II? While such information should not lead to disregarding of inequality, it would all the same nuance the picture.

Literature suggests that children with milder CP are more responsive to early motor interventions than children with more severe forms of CP (Morgan et al., 2018; Sakzewski et al., 2019). So, in contrast with reviewer 2s suggestion, children with milder motor impairments would benefit more from early intervention. Our experience from clinical practice is that children with less severe motor impairments do show aberrant motor development, but this does not necessarily mean that the child is delayed. We have added a sentence with references addressing that children with milder CP are more responsive to early intervention in the first section of the introduction. We have also added a section in the discussion addressing that children with milder CP and especially those with unilateral CP might have near to normal motor milestones acquisition and the need to assess quality of movement to enable early identification of infants with CP. 

Method

• 72-74: Exclusion criteria seems to exclude all children not born in Uppsala regardless of when they moved to Uppsala? Please clarify the exclusion criteria and the reasons for it as excluded children correspond to almost 50% of your current sample

As the aim of our study was to investigate children’s access to early physiotherapy, i.e. during the first months of life, children with CP who moved into the county after 1 year of age were excluded from the study. We have added this to the text.

• 78-79: Do all term born children always participate in the well child program?

Yes, the Swedish child health services are offered to all children aged 0-5 years and has an attendance rate of 97%. We have added this in the introduction. 

• 79-80: Please clarify which children receives neonatal follow-up and how neonatologist differs from neuropediatrician. According to line 52-53, neonatal follow-up would not only include <32 weeks. Do the preterm children also participate in the well child program? 82: The neuropediatrician seem to be part of the neonatal follow-up, which further complicates the relevance of this division of referral source, see lines 52 and 79.

We have removed the national guideline to follow-up infants born before 28 weeks of gestation when describing neonatal follow-up in the introduction, and instead explained the inclusion criteria for neonatal follow-up in Uppsala County in the “study population and setting” section. All children are offered the well-child surveillance, and CHS reaches 97% of all Swedish children 0-5 years of age. We have rewritten this section of the introduction to make this more concise. 

The neonatologist work with newborn infants that are in need of neonatal intensive care. Given that these children are at increased risk for later developmental disorders these high-risk infants are routinely enrolled in neonatal follow-up programs. In Uppsala County, the neonatologist is the responsible physician at the follow-up. If a neurological impairment or disability is suspected the child is referred to a neurologist, which work at another department and is not a part of the neonatal follow-up. As we have changed our aim, research questions and results, this step in the children’s clinical journey is no longer investigated in our study and we have removed this from the text. 

After redefining our aim, we divided the children into those who had been referred by the neonatologist or enrolled in the neonatal follow-up and those referred from other healthcare professionals. Hopefully this division makes it clearer. 

• 102: Please consider if multilevel analysis would be useful, e.g. multivariable logistic regressions.

We added a univariate and multivariate regression analyses as suggested.

• 109, Ethics: I disagree with your statement that all data are presented on group level. Some groups consists of single children. In a fairly small community of children with CP and their parents, in a named region in Sweden, there would be a significant risk that individual children are identified by some of these data. This would be particularly unfortunate since no informed consent is collected.

We agree and to avoid identifications of individual children, we have collapsed some variables to have more children in every groups. 

Results

• 117-118: Please clarify the meaning of “missed early” and why this is reason for exclusion.

We have rewritten this sentence and added another to provide more clarity. However, we cannot describe this in more detail as it could lead to identification of this child. 

• Table 1: In the rows you explain that you report n (%), however the percentages are summed up by column. I would recommend you to reconsider this way of presenting in order to gain readability

We agree and have edited this in table 1.

• 120-121: Please explain the relevance of the difference between being referred by neonatologist or neuropediatrician – are these children not part of the same neonatal follow-up? Also, is it possible that CHS does not refer to physiotherapist directly, but via e.g. neuropediatrician, thus contributing to the delay?

The neonatologist work with newborn infants that are in need of neonatal intensive care. Given that these children are at increased risk for later developmental disorders these high-risk infants are routinely enrolled in neonatal follow-up programs. In Uppsala County, the neonatologist is the responsible physician at the follow-up. If a neurological impairment or disability is suspected the child is referred to a neurologist, which work at another department and is not a part of the neonatal follow-up. As we have changed our aim, research questions and results, this step in the children’s clinical journey is no longer investigated in our study and we have removed this from the text. 

In regards to referral from the CHS to the pediatric neurologist. Linear serial models of referrals are known to delay access to intervention for children with CP, so this is definitively a warranted concern. In our sample, all children were referred to the neurologist due to other medical concerns, such as seizures. However, after redefining our aim, we divided the children into those referred by neonatologists or enrolled in neonatal follow-up and those referred from other healthcare professional this is no longer of importance as they now are presented as one group. We have added a sentence in the discussion that linear serial model of referral should be avoided as it can prolong lead time and delay intervention. 

• Table 2: It would be helpful if you explain H and Df among the abbreviations in the table and also elaborate the table heading so that the table can be understood separately from the text. GMFCS is quite close to significant regarding first visit to physiotherapy – I propose you add to the discussion whether your results actually show that there is no difference with regard to GMFCS or if the results may be due to power-issues.

We have removed table 2 from the manuscript and added a discussion in ‘Limitations’ regard sample size. 

Discussion

• 169: It is a bit unclear if you refer to the Ref 12 or to own results (or both)

We are referring to Ref 12 and have rewritten this sentence to convey the correct meaning. 

• 172: As mentioned above, I find it a difficult to follow this grouping of children due to referral source – which are the children who are not from CHS and not in the neonatal follow-up. There is a lack of consistency and/or a need of a more concise description throughout the manuscript.

After redefining our aim, we divided the children into those referred by a neonatologist or enrolled in neonatal follow-up or those referred by other healthcare professionals. Hopefully this division makes the groups easier to follow when reading the manuscript. 

• 175-176: Again, as mentioned earlier, I would suggest you elaborate around this matter. While it is important to provide equal care, as to receive early FT assessment, there may also be limited consequences for children with mild motor difficulties to be referred to physiotherapy at a higher age, because of their fairly good motor function(?). Other difficulties associated with CP may be prioritized when motor function is good, such as e.g. speech or cognition.

All children should have equal access to physiotherapy/early motor interventions. Knowing that milder forms of CP are more responsive to early motor interventions (Morgan et al., 2018; Sakzewski et al., 2019), stress the importance of identifying these children early. At this point we are missing the children where our intervention could have the largest effect. Delayed access to physiotherapy can lead to delayed motor development as well as muscle weakness, contractures and loss of function (Ulrich, 2010). Furthermore, as motor abilities are needed to explore the world, solve problems and make discoveries (Johnston, 2009; Von Hofsten, 2004), early motor problems are likely to affect child development. Already from an early age these motor behaviors and the children’s perceptual-motor experiences within cultural and social contexts form their cognition (Johnston, 2009; Von Hofsten, 2004). Moreover, as neuronal plasticity is enhanced during the first months of life and goal-directed actions lead to structural and functional changes in the brain motor interventions can impact how the brain organizes itself (Johnston, 2009; Novak et al., 2017). This is likely why early motor interventions programs have shown to be beneficial for children’s cognitive outcomes (Hadders-Algra, 2021; Morgan et al., 2016).

Due to the course of development of speech, early intervention in this regards is not relevant during the first months of a child’s life. 

• 178: Please clarify if the early interventions refer to physiotherapy or other interventions. 

We have removed this section of the discussion. 

• Please consider discussion about power in your study when discussing GMFCS as not significantly associated with early referral to physiotherapist.

We have discussed this in the ‘Limitations’ section. 

• 205-206: It seems in this statement like all children are not included in the neonatal follow-up – is that right and if so, please state this more clearly. According to the introduction line 52 all children with complicated birth history are routinely included.

We have deleted this section of the discussion.

• 206: Please specify which interventions and which guidelines you refer to.

We have rewritten this section and added the name of the guidelines we refer to.

• 216: Please specify the kind of review you refer to.

We have rewritten this sentences to convey the correct meaning. 

• 220-232: To implement use of evidence based motor assessment tools within CHS seems to be a suitable suggestion in order to early detect more children with presumed CP. I suggest you emphasize this even more. (As there is a shortage of specialist CHS nurses in primary care, maybe such assessment could be introduced as a cooperation task for primary health care physiotherapists in order to widen their area of responsibility and thereby strengthening their professional role?)

When rewriting the discussion we emphasize CHS as the arena where we can do most to improve timely identification of children with CP. We also highlight the benefits and challenges of using evidence-based assessment methods in well-child surveillance. In Sweden, child health nurses outnumber pediatric physiotherapists. Additionally, the CHS reaches 97% of all children aged 0-5 years of age, while physiotherapist only see children on a referral-basis. By introducing an evidence-based assessment method and training of child health nurses, we would potentially be able to reduce the unequal access for early intervention. This holds not only for CP, but for other developmental disorders as well. As we all realize the unleashed potential of the CHS as an arena for early identification of children with motor problems, there are continuously quality improvement projects to increase the cooperation between child health nurses and pediatric physiotherapists. 

Sincerely,

Kine Johansen

Physiotherapist specialised in paediatrics, PhD

Corresponding author 

Department of Women’s and Children’s Health

References

Hadders-Algra, M. (2021). Early Diagnostics and Early Intervention in Neurodevelopmental Disorders—Age-Dependent Challenges and Opportunities. Journal of Clinical Medicine, 10(4), 861. https://doi.org/10.3390/jcm10040861

Johnston, M. V. (2009). Plasticity in the developing brain: implications for rehabilitation. In Developmental Disabilities Research Reviews (Vol. 15, Issue 2, pp. 94–101). John Wiley and Sons Inc. https://doi.org/10.1002/ddrr.64

Morgan, C., Darrah, J., Gordon, A. M., Harbourne, R., Spittle, A., Johnson, R., & Fetters, L. (2016). Effectiveness of motor interventions in infants with cerebral palsy: a systematic review. Developmental Medicine & Child Neurology, 58(9), 900–909. https://doi.org/10.1111/dmcn.13105

Morgan, C., Fahey, M., Roy, B., & Novak, I. (2018). Diagnosing cerebral palsy in full-term infants. Journal of Paediatrics and Child Health, 54(10), 1159–1164. https://doi.org/10.1111/jpc.14177

Novak, I., Morgan, C., Adde, L., Blackman, J., Boyd, R. N., Brunstrom-Hernandez, J., Cioni, G., Damiano, D., Darrah, J., Eliasson, A. C., De Vries, L. S., Einspieler, C., Fahey, M., Fehlings, D., Ferriero, D. M., Fetters, L., Fiori, S., Forssberg, H., Gordon, A. M., … Badawi, N. (2017). Early, accurate diagnosis and early intervention in cerebral palsy: Advances in diagnosis and treatment. In JAMA Pediatrics (Vol. 171, Issue 9, pp. 897–907). American Medical Association. https://doi.org/10.1001/jamapediatrics.2017.1689

Novak, I., Morgan, C., Fahey, M., Finch-Edmondson, M., Galea, C., Hines, A., Langdon, K., Namara, M. M., Paton, M. C., Popat, H., Shore, B., Khamis, A., Stanton, E., Finemore, O. P., Tricks, A., te Velde, A., Dark, L., Morton, N., & Badawi, N. (2020). State of the Evidence Traffic Lights 2019: Systematic Review of Interventions for Preventing and Treating Children with Cerebral Palsy. In Current Neurology and Neuroscience Reports (Vol. 20, Issue 2, pp. 1–21). Springer. https://doi.org/10.1007/s11910-020-1022-z

Sakzewski, L., Sicola, E., Verhage, C. H., Sgandurra, G., & Eliasson, A. C. (2019). Development of hand function during the first year of life in children with unilateral cerebral palsy. Developmental Medicine and Child Neurology, 61(5), 563–569. https://doi.org/10.1111/dmcn.14091

Ulrich, B. D. (2010). Opportunities for early intervention based on theory, basic neuroscience, and clinical science. Physical Therapy, 90(12), 1868–1880. https://doi.org/10.2522/ptj.20100040

Von Hofsten, C. (2004). An action perspective on motor development. In Trends in Cognitive Sciences (Vol. 8, Issue 6, pp. 266–272). Trends Cogn Sci. https://doi.org/10.1016/j.tics.2004.04.002

---

## [Decision Letter · Decision Letter 1]

5 May 2021

PONE-D-20-28394R1

Early access to physiotherapy for infants with Cerebral Palsy: a retrospective chart review

PLOS ONE

Dear Dr. Johansen,

Thank you for submitting your manuscript to PLOS ONE. After careful consideration, we feel that it has merit but does not fully meet PLOS ONE’s publication criteria as it currently stands. Therefore, we invite you to submit a revised version of the manuscript that addresses the points raised during the review process.

We look forward to receiving your revised manuscript.

Kind regards,

Christos Papadelis, Ph.D.

Academic Editor

PLOS ONE

Journal Requirements:

Reviewers' comments:

Reviewer's Responses to Questions

**Comments to the Author**

1. If the authors have adequately addressed your comments raised in a previous round of review and you feel that this manuscript is now acceptable for publication, you may indicate that here to bypass the “Comments to the Author” section, enter your conflict of interest statement in the “Confidential to Editor” section, and submit your "Accept" recommendation.

Reviewer #1: All comments have been addressed

Reviewer #2: All comments have been addressed

2. Is the manuscript technically sound, and do the data support the conclusions?

Reviewer #1: Yes

Reviewer #2: Yes

3. Has the statistical analysis been performed appropriately and rigorously? 

Reviewer #1: Yes

Reviewer #2: Yes

4. Have the authors made all data underlying the findings in their manuscript fully available?

Reviewer #1: No

Reviewer #2: No

5. Is the manuscript presented in an intelligible fashion and written in standard English?

Reviewer #1: Yes

Reviewer #2: Yes

6. Review Comments to the Author

Reviewer #1: The authors have revised the manuscript thoroughly, and overall revisions are clear and logical. A main point I still have is on the regression analysis. Firstly, the authors did not state the regression models they used are linear or nonlinear. Secondly, the names of regressions the authors used, univariate regression and multivariate regression, are confusing. In both regression analysis, there is only one outcome variable (age at the first visit to a physiotherapist), thus the regressions should be named as simple (linear, I assume) regression for that with only one predictor and multiple (linear, I assume) regression for that with more than one predictors. Thirdly, as to the multivariate regression, the authors did not include an interaction item on referral source * number of risk factors. As mentioned before, I think there may exist interaction effect between referral source and number of risk factors. I am interested in that if the interaction item on referral source * number of risk factors was added in the multivariate regression, how would the results be?

Reviewer #2: Thank you for the opportunity to re-review this manuscript. Major revisions are made to the manuscript, which greatly have improved it's quality, readability and arguments. All my queries are addressed. I have no further comments.

7. PLOS authors have the option to publish the peer review history of their article (what does this mean?). If published, this will include your full peer review and any attached files.

Reviewer #1: **Yes: **Yanlong Song

Reviewer #2: No

---

## [Author Response · Author response to Decision Letter 1]

6 Jun 2021

Response to reviewers comments regarding manuscript PONE-D-20-28394R2

We would again like to thank the reviewers for their valuable and constructive feedback! Below you find our response to the queries. We have highlighted the changes in the manuscript in yellow. Added text is highlighted, while removed text appears as strikethrough.

Journal Requirements

(1) Please review your reference list. 

We have reviewed the references and it is complete and correct. Small changes have been made in the text. Additionally, we have replaced reference 6, with a recently published paper (reference 8).

Removed

Reference 6. Morgan C, Darrah J, Gordon AM, Harbourne R, Spittle A, Johnson R, et al. Effectiveness of motor interventions in infants with cerebral palsy: a systematic review. Dev Med Child Neurol. 2016;58: 900–909. doi:10.1111/dmcn.13105. 

Replaced

Reference 8. Morgan C, Fetters L, Adde L, Badawi N, Bancale A, Boyd RN, et al. Early intervention for children aged 0 to 2 years with or at high risk of cerebral palsy: international clinical practice guideline based on systematic reviews. JAMA Pediatr. 2021. doi: 10.1001/jamapediatrics.2021.0878. 

Review comments to the Author

Reviewer #1

(1) The authors have revised the manuscript thoroughly, and overall revisions are clear and logical. A main point I still have is on the regression analysis. Firstly, the authors did not state the regression models they used are linear or nonlinear. Secondly, the names of regressions the authors used, univariate regression and multivariate regression, are confusing. In both regression analysis, there is only one outcome variable (age at the first visit to a physiotherapist), thus the regressions should be named as simple (linear, I assume) regression for that with only one predictor and multiple (linear, I assume) regression for that with more than one predictors. Thirdly, as to the multivariate regression, the authors did not include an interaction item on referral source * number of risk factors. As mentioned before, I think there may exist interaction effect between referral source and number of risk factors. I am interested in that if the interaction item on referral source * number of risk factors was added in the multivariate regression, how would the results be?

As reviewer #1 assumed, we have used a linear regression model. We have added this to the text. To further clarify our analysis, we have changed the wording from univariate and multivariate to unadjusted and adjusted regression models. For the third point, we will start by thanking the reviewer! This was important and has added valuable information about our sample and our results. We have included the interaction item in the regression analysis and added a figure to illustrate the interaction between referral source and number of risk factors. Text addressing these results have be added where appropriate. Additionally, we have changed the order of the variable “referral source”, so that the NICU graduates are used as the reference. 

Sincerely,

Kine Johansen

Physiotherapist specialised in paediatrics, PhD

Corresponding author 

Department of Women’s and Children’s Health

---

## [Editor Report · Decision Letter 2]

15 Jun 2021

Early access to physiotherapy for infants with cerebral palsy: a retrospective chart review

PONE-D-20-28394R2

Dear Dr. Johansen,

We’re pleased to inform you that your manuscript has been judged scientifically suitable for publication and will be formally accepted for publication once it meets all outstanding technical requirements.

Kind regards,

Christos Papadelis, Ph.D.

Academic Editor

PLOS ONE
---

## [Editor Report · Acceptance letter]

17 Jun 2021

PONE-D-20-28394R2 

Early access to physiotherapy for infants with cerebral palsy: a retrospective chart review 

Dear Dr. Johansen:

I'm pleased to inform you that your manuscript has been deemed suitable for publication in PLOS ONE. Congratulations! Your manuscript is now with our production department. 

Kind regards, 

on behalf of

Dr. Christos Papadelis 

Academic Editor

PLOS ONE